# AA-amyloidosis in cats (*Felis catus*) housed in shelters

Filippo Ferri[1,2,3], Silvia Ferro[4]*, Federico Porporato[1,2], Carolina Callegari[1], Chiara Guglielmetti[5], Maria Mazza[5], Marta Ferrero[5], Chiara Crinò[6], Enrico Gallo[4], Michele Drigo[3], Luigi Michele Coppola[3], Gabriele Gerardi[3], Tim Paul Schulte[7], Stefano Ricagno[7,8], Monique Vogel[9,10], Federico Storni[9,10,11], Martin F. Bachmann[9,10], Anne-Cathrine Vogt[9,10], Serena Caminito[12], Giulia Mazzini[13], Francesca Lavatelli[12], Giovanni Palladini[13], Giampaolo Merlini[13], Eric Zini[1,3,14]

1 AniCura Istituto Veterinario di Novara, Granozzo con Monticello, Novara, Italy, 2 Studio Veterinario Associato Vet2Vet di Ferri e Porporato, Orbassano, Torino, Italy, 3 Department of Animal Medicine, Production and Health, University of Padova, Legnaro, Padova, Italy, 4 Department of Comparative Biomedicine and Food Sciences, University of Padova, Legnaro, Padova, Italy, 5 Istituto Zooprofilattico Sperimentale del Piemonte, Liguria e Valle d'Aosta, SC Diagnostica Specialistica, Torino, Italy, 6 Department of Clinical Science and Services, The Royal Veterinary College, Hatfield, United Kingdom, 7 Institute of Molecular and Translational Cardiology, IRCCS Policlinico San Donato, Milan, Italy, 8 Departments of Biosciences, La Statale, University of Milan, Milan, Italy, 9 Department of Rheumatology and Immunology, University Hospital Bern, Bern, Switzerland, 10 Department of BioMedical Research, University of Bern, Bern, Switzerland, 11 Department of Visceral Surgery and Medicine, University Hospital of Bern, Bern, Switzerland, 12 Department of Molecular Medicine, University of Pavia, Pavia, Italy, 13 Amyloidosis Research and Treatment Center, Fondazione IRCCS Policlinico San Matteo and University of Pavia, Pavia, Italy, 14 Clinic for Small Animal Internal Medicine, Vetsuisse Faculty, University of Zurich, Zurich, Switzerland

* silvia.ferro@unipd.it

**Data Availability Statement:** All relevant data are within the paper and its Supporting information files.

**Funding:** Eric Zini: BIRD 193233 University of Padova; 2019 and 2020 Research Funds AniCura.

## Abstract

Systemic AA-amyloidosis is a protein-misfolding disease characterized by fibril deposition of serum amyloid-A protein (SAA) in several organs in humans and many animal species. Fibril deposits originate from abnormally high serum levels of SAA during chronic inflammation. A high prevalence of AA-amyloidosis has been reported in captive cheetahs and a horizontal transmission has been proposed. In domestic cats, AA-amyloidosis has been mainly described in predisposed breeds but only rarely reported in domestic short-hair cats. Aims of the study were to determine AA-amyloidosis prevalence in dead shelter cats. Liver, kidney, spleen and bile were collected at death in cats from 3 shelters. AA-amyloidosis was scored. Shedding of amyloid fibrils was investigated with western blot in bile and scored. Descriptive statistics were calculated. In the three shelters investigated, prevalence of AA-amyloidosis was 57.1% (16/28 cats), 73.0% (19/26) and 52.0% (13/25), respectively. In 72.9% of cats (35 in total) three organs were affected concurrently. Histopathology and immunofluorescence of post-mortem extracted deposits identified SAA as the major protein source. The duration of stay in the shelters was positively associated with a histological score of AA-amyloidosis (B = 0.026, CI95% = 0.007–0.046; p = 0.010). AA-amyloidosis was very frequent in shelter cats. Presence of SAA fragments in bile secretions raises the possibility of fecal-oral transmission of the disease. In conclusion, AA-amyloidosis was very frequent in shelter cats and those staying longer had more deposits. The cat may represent a natural model of AA-amyloidosis.

Maria Mazza: 10/19 RC Istituto Zooprofilattico
Sperimentale del Piemonte Liguria e Valle d'Aosta.
Stefano Ricagno: this study was partially supported
by Ricerca Corrente funding from Italian Ministry of
Health to IRCCS Policlinico San Donato. The
funders had no role in study design, data collection
and analysis, decision to publish, or preparation of
the manuscript.

**Competing interests:** The authors have declared
that no competing interests exist.

## Introduction

Amyloidoses are classified according to the fibril protein precursor and the distribution of amyloid deposition. More than 30 amyloid proteins have been identified in humans and 10 have been documented in animals [1–3]. AA-amyloidosis, the most commonly reported form in animals, is derived from the deposition of serum amyloid-A protein (SAA), an apolipoprotein produced in the liver upon stimulation by proinflammatory cytokines during inflammatory or neoplastic disorders. This form of AA-amyloidosis is referred to as reactive [4–7]. In companion animals, deposition of AA-amyloid occurs in the familial form in specific breeds, including Abyssinian and Siamese cats and Shar-Pei dogs [3, 8].

In cats the exact pathogenesis of AA-amyloidosis has not been fully clarified. However, several hypotheses have been proposed, such as increased circulating concentrations of SAA along with defects in the degrading properties of monocytes or genetic structural abnormalities of proteins [3]. Many conditions, including neoplastic, inflammatory, and metabolic diseases are associated with high production of SAA in cats [9] but amyloidosis has been only sporadically reported in not predisposed breeds [10–13]; recently, the differences in SAA genetic sequences proved to not contribute to the pathogenesis of feline AA-amyloidosis both in Japanese domestic cats [14] and in Abyssinian breed [15]. Other factors, such as amyloid-enhancing factors and glycosaminoglycans, may possibly take part in amyloidogenesis by accelerating amyloid-fibril deposition [16].

The transmissibility of AA-amyloidosis has been demonstrated in some animal species as taking place through a prion-like seeding-nucleation-dependent mechanism [17]. In addition, in mice the development of AA-amyloidosis induced by injection of an inflammatory stimulant appears to be accelerated by the oral administration of semi-purified amyloid fibrils extracted from murine, humans and cattle affected tissues, leading to hypothesize intra-species and cross-species transmissibility [18]. The presence of AA-amyloids in feces from cheetahs pointed to a potential oral-fecal transmission route, possibly justifying the high prevalence of the disease in this species [7].

In animals, the organs most commonly involved are the spleen, liver, and kidneys, as well as gastrointestinal mucosa [3, 6]. Clinical signs depend on the localization and amount of the deposition and reflect the extent of organ damage. Chronic kidney disease is common in affected Abyssinian cats and Shar-Pei dogs, whereas hepatic dysfunction has been reported in Siamese cats with liver deposits. Several studies have described AA-amyloidosis in Siamese and Abyssinian cats, while it has rarely been reported in domestic short-hair cats [10–13]. Amyloidosis was reported in one third of cats naturally infected with feline immunodeficiency virus (FIV), but not in those experimentally infected [19].

The present study was initiated after the observation that three cats from the same shelter were diagnosed with systemic AA-amyloidosis within six months. The aims of this study were to investigate the prevalence of AA-amyloidosis in domestic short-hair cats housed in shelters at death, describe clinical and histopathology findings, and assess the possible biliary excretion of SAA fragments.

## Materials and methods

### Cats and clinical data

Seventy-nine cats were prospectively enrolled from three shelters in Northern Italy, each approximately 100 km apart (shelters A, B, and C). Cats were included if they had died spontaneously or had been euthanized due to the severity of their clinical condition between January and September 2019, and if tissue samples had been collected within six hours from death, not

allowing for a full necropsy to be recorded. Euthanasia was performed according to American Veterinary Medical Association (AVMA) Guidelines for the Euthanasia of Animals (2020). Data regarding gender, age at admission to the shelter, length of stay, cause of death, age at death, and presence of concurrent chronic diseases were obtained from medical records. Data about the cause of death were available for 59 cats and about the presence of concurrent chronic diseases for 22.

## Mice and antisera preparation

Female BALB/cOlaHsd were used (Envigo, Amsterdam, The Netherlands). All animals were acclimatized to the facility for at least one week before experiments were performed. Experiments were approved by the animal ethic committee of the Swiss Cantonal Veterinary Office (Permission nr: BE70/18). For blood collection mice were terminally bled by cardiac puncture, under deep anaesthesia. Antigenic preparations from the N-terminal region (MREANYIGAD), the central region (QRGPGGAWAAKV) and the C-terminal region (EWGRSGKDPNHFRP) of feline serum SAA were synthesized and coupled with Qβ virus-like particles (VLPs) [20]. Each VLP-peptide at a concentration of 30 mg/ml was administered once to BALB/c mice. Fourteen days after immunization, mice were bled and the specificity of antisera against each peptide was assessed by ELISA using each synthetic peptide.

## Histology and immunofluorescence

Liver, spleen and kidney samples were collected and fixed in 10% buffered formalin for at least 24 hours and then embedded in paraffin. To identify amyloid deposits, 4 µm thick tissue sections were stained with hematoxylin and eosin (HE) for examination under standard light microscopy, and with Congo red for examination under standard and polarized light microscopy [21]; staining with Congo red is a qualitative method used for the identification of amyloids in vitro and in tissue sections [4]. A semi-quantitative histological score from 0 to 2 was applied based on the amount of Congo red-positive material observed under light microscopy and confirmed under polarized light.

A score of 0 was assigned to samples without positivity, while scores of 1 and 2 were assigned in the case of mild and moderate-to-marked amounts, respectively. To estimate the overall amount of tissue deposits in each affected cat, an amyloidosis score (from 0 to 6) was created by adding the histological score of the liver, spleen and kidney.

Immunofluorescence was performed on feline liver, spleen, and kidney samples to confirm AA-amyloidosis. Each staining was performed as previously described [22]. Briefly, 1% thioflavine S (T1892, Sigma-Aldrich, Buchs, Switzerland) in ddH$_2$O was used to stain for amyloid aggregates directly after deparaffinization, and nuclei were stained with DAPI. To examine the interaction of amyloid deposits with antiserum, serum from vaccinated mice (1:10 dilution) was added, followed by a rat anti-mouse IgG conjugated to biotin (Cat#13-4013-85, Thermo-Fisher, Basel, Switzerland) and a streptavidin conjugated Alexa546 (s11225, Molecular Probes, Eugene, OR, USA). Pictures were acquired with AxioImager A2 and AxioCam (Carl Zeiss, Jena, Germany). Pancreatic tissue sample from a cat without AA-amyloidosis was used as negative control (S1 Fig) and kidney with AA-amyloidosis as positive control (S2 Fig).

## Infectious diseases in cats

Three slices, 15 µm each, were obtained from formalin-fixed paraffin-embedded splenic samples, and RNA was extracted (High Pure RNA, Roche Diagnostics, Mannheim, Germany). RT-qPCR was used to assess the presence of FIV, feline leukemia virus (FeLV) and feline coronavirus (FCoV). The LightCycler 96 (Roche Diagnostics) was used for amplifications with the

Oasig lyophilized OneStep RT-qPCR Master Mix along with three different commercial assays: Feline Immunodeficiency Virus Standard Kit, Feline Leukemia Virus Standard Kit and Feline Coronavirus Advanced Kit (Genesig, Chandler's Ford, UK). The manufacturers' instructions were followed for reagents and thermic protocols. Assay performance was assessed by testing 10-fold dilutions of the external positive control (declared concentration of $2x10^5$ viral copies), with a limit of detection (LOD) of $2x10^1$, $2x10^0$, $2x10^0$ and an efficiency of 2.01, 1.87, and 1.88 for the quantification of FIV, FeLV, and FCoV, respectively. Samples were considered positive with Ct<40.

## SAA fragments in bile

Samples of bile were collected from the gallbladder of deceased cats. Two techniques (Congo red staining, western blotting) were used to investigate the possible presence of SAA protein or amyloid fibrils in cats' bile. Briefly, 20 volumes of 0.15 M NaCl were added to 150 μg of sample, followed by homogenization for 1 min. After centrifugation at 9418 rcf for 15 min at 20 ˚C, the supernatants were discarded and pellets dissolved in 20 volumes of 0.15 M NaCl, and then homogenized and centrifuged. This washing step was repeated twice. The pellets were dissolved in 300 μl of ultrapure water, and homogenized and centrifuged at 40,500 rcf for 1 hr at 10 ˚C. The supernatants were discarded and the pellets were weighed. Ten volumes of ultrapure water were added and, after homogenization, samples were centrifuged at 215,000 rcf for 90 min at 10 ˚C. Pellets were finally resuspended in 30 μl of ultrapure water. Samples were quantified (Qubit Protein Assay kit, Invitrogen) according to the manufacturer's instructions. For western blotting, 30 μg of protein extracts were loaded in each well for the analysis. On 24 samples, Laemmli buffer (2x) was added to each protein sample, followed by SDS-PAGE electrophoresis and transferred onto PVDF membranes (Immobilon-P, Millipore, Billerica, MA, USA). Blots were blocked by Tris-buffered saline-bovine serum albumin 1% and incubated with Rabbit pAb SAA1 (AB171030, Abcam, Milan, Italy) diluted 1:1000 in 0.1% Tween-20/Tris-buffered saline. Immunodetection was carried out with an alkaline phosphatase-conjugated anti-rabbit immunoglobulin (Vector Laboratories, Peterborough, UK) diluted 1:15000 in 0.1% Tween-20/Tris-buffered saline and revealed by a chemiluminescent substrate. The images of the blots were captured by a Chemidoc Touch system (Bio-Rad, Hercules, CA, USA).

Finally, a numerical score was arbitrarily attributed to the results of the western blot, based on the degree of positivity of each sample: 0 for negative samples, 1+, 2+, and 3+ for mild, moderate, and severe positivity, respectively. In addition, the pellets from the bile of 3 cats with liver amyloidosis and displaying positive SAA signal on western blotting were evaluated by Congo red staining, in order to detect possible amyloid fibrils. For Congo red staining, aliquots of the pellet were deposited onto glass slides and dried. Congo red staining was performed using alkaline Congo red–saturated alcoholic solution as previously described [23, 24].

## Statistical analysis

Descriptive statistics (median and interquartile interval, i.e., Q1-Q3) were calculated for numerical continuous variables including age, duration of stay in the shelter, and AA-amyloidosis additive score. Frequencies (counts and percentages) were calculated for gender and AA-amyloidosis status (positive or negative), both for the shelter and the entire population of sampled cats. Frequencies of positivity for AA-amyloidosis in the histological samples and for western blot in bile samples were also calculated. Differences between shelters and between cats with and without AA-amyloidosis were analysed for continuous variables with Kruskall-Wallis and Mann-Whitney tests followed by Bonferroni adjustment. For qualitative

variables, rxc contingency tables, chi-square tests with Yate's correction, or Fisher's exact tests were used.

In a subset of cats with available medical records and follow-up (n = 20, all belonging to shelter C), a disease duration index was calculated. The index represented the proportion of time a cat was sick during its stay in the shelter, irrespective of the disease and severity, and ranged from 0 to 1. The independent sample non-parametric t-test was used to assess the difference in the median disease duration index between cats with and without AA-amyloidosis.

Finally, multivariable analysis that included the relevant predictors for the biological model or the univariate screening with a tolerant p-value <0.15 was used to assess the effects of shelter, age, and duration of stay on the presence of AA-amyloidosis. Two multivariable models were built: a multivariable logistic regression model (S1 Model) to investigate AA-amyloidosis as a dichotomous variable (positive/negative) and a generalized linear model (S2 Model) to investigate the severity of AA-amyloidosis deposition based on the AA-amyloidosis additive score. The Hosmer-Lemeshow goodness of fit for S1 Model and Levene's test of equality of error variance for S2 Model, were applied. In both cases, the interaction term between shelter and age or duration of stay was initially included, but excluded from the final models if not significant. The level of significance was set at p = 0.05. Statistical analyses were carried out using SPSS v. 26.0 (IBM, Armonk, NY, USA).

## Results

### Cats and clinical data

Clinical and histopathological data were collected post-mortem from 79 domestic short-hair cats that deceased in three independent shelters 100 km apart from each other in Northern Italy. Twenty-eight (35.4%), 26 (32.9%), and 25 (31.7%) of the investigated domestic short-hair cats were from shelters A, B, and C, respectively. The cats stayed in the shelters for a median duration of 27 months (Q1-Q3: 9–64) and median age at death was 6 years (Q1-Q3: 4–8). Cats were younger in shelter A compared to B (p = 0.009) and C (p = 0.026), while the duration of stay was shorter in shelter A compared to B (p = 0.021), but not to C (p = 0.093). Gender distribution was similar among shelters (Table 1, S1 Table). While 12 (15.2%) cats were euthanized due to worsening clinical conditions, 67 cats (84.8%) died spontaneously. Fourteen cats died as a consequence of kidney failure among 47 cats (29.8%) for which the cause of death was reported (S1 Table). Information on other chronic diseases was obtained only for 22 cats from shelter C; 18 had one or more chronic diseases, the most common of which were feline leukemia virus infection, chronic enteropathy, gingivostomatitis, and respiratory tract disease (S1 Table).

### Histology and immunofluorescence

Overall, 38 (48.1%) liver, 46 (58.2%) spleen, and 40 (50.6%) kidney samples of cats (Table 2) had amyloid deposits, with 48 animals (60.8%) showing at least one of the three organs affected

**Table 1. Gender distribution, age and length of stay of the 79 cats in the three shelters.**

| Cat-shelter | Females N (%) | Males N (%) | Age in years, median (Q1-Q3) | Length of stay in months, median (Q1-Q3) |
|---|---|---|---|---|
| Shelter A | 14 (50) | 14 (50) | 5 (4–6) | 14 (8–28) |
| Shelter B | 10 (38.5) | 16 (61.5) | 7 (5–10) | 38 (17–72) |
| Shelter C | 12 (48) | 13 (52) | 7 (4–10) | 32 (13–84) |

N = number; % = percentage

**Table 2. Histological score of liver, spleen, and kidney samples positive to Congo red staining and showing birefringence under polarized light microscopy in in 79 cats.**

| Histological score | Kidney N (%) | Liver N (%) | Spleen N (%) |
|---|---|---|---|
| 0 (negative) | 39 (49.4) | 41 (51.9) | 33 (41.8) |
| 1 (moderate) | 16 (20.2) | 24 (30.4) | 10 (12.7) |
| 2 (marked) | 24 (30.4) | 14 (17.7) | 36 (45.5) |
| Total | 79 (100) | 79 (100) | 79 (100) |

N = number; % = percentage

(S1 Table). Of the affected cats, 35 (72.9%) had three concurrently involved organs, 6 (12.5%) had two organs, and 7 (14.6%) had one organ (5 spleen, 1 liver and 1 kidney, respectively) (S1 Table). The protein SAA was identified within fibril deposits by immunofluorescence. Custom antibodies raised in mice recognized amyloid deposits in the liver (Fig 1a), kidney (Fig 1d), and spleen (Fig 1g), confirming AA-amyloidosis in all affected cats.

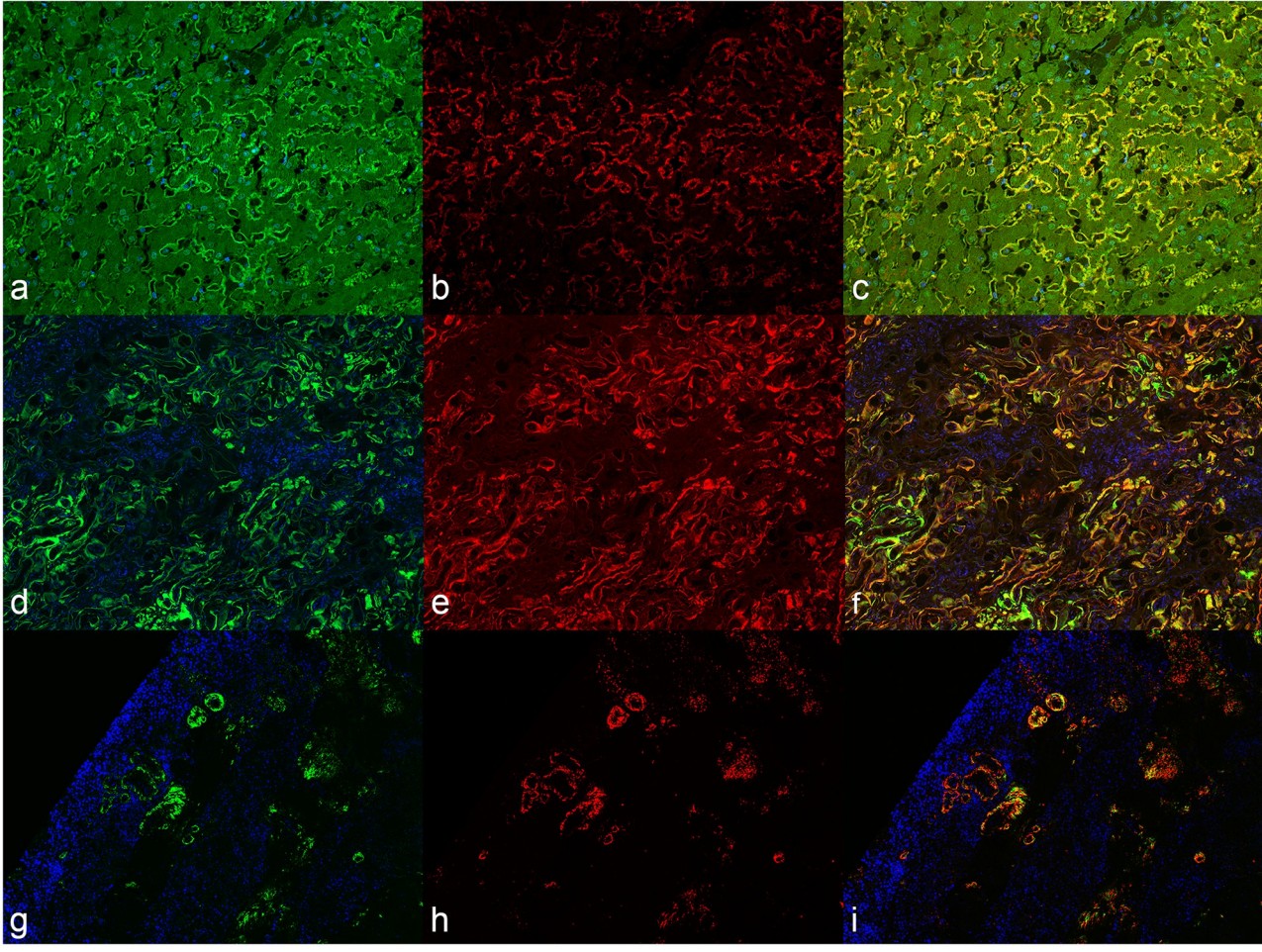

**Fig 1. Amyloidosis, cat, fluorescence-immunohistochemistry for AA-amyloidosis.** Tissue were stained with Thioflavine S (ThioS) to confirm the presence of amyloidogenic aggregates [a, d, g, left column, ThioS in green, 4',6-diamidino-2-phenylindole (DAPI) in blue] and with an IgG derived from mice immunized with Qβ-AA peptide vaccine (b, e, h, middle column, IgG in red). A co-localization staining of ThioS and IgG (c, f, i, right column, merge) was generated to authenticate the specificity of the IgGs. a, b, c) Liver. d, e, f) Kidney. g, h, i) Spleen.

Out of the 48 positive cats for AA amyloidosis, 16 (33.3%), 19 (39.6%), and 13 (27.1%) belonged to shelter A, B, and C, respectively. The post-mortem prevalence of AA-amyloidosis was 57.1% (16/28 cats), 73.0% (19/26), and 52.0% (13/25), respectively. The overall prevalence of AA-amyloidosis as well as distribution of the affected organ was not significantly different between shelters. Of the 22 cats with available information on other diseases, 12 showed AA-amyloidosis in at least one of the three organs with 11 cats (91.7%) being affected by one or more concurrent chronic diseases and one cat (8.3%) with no concurrent chronic disease. The AA-amyloidosis histological score (Table 2) of the spleen was higher than the two other organs (p<0.001). The scores did not differ between liver and kidney, nor between shelters. The distribution of the AA-amyloidosis additive score used in multivariable analysis is reported in (S2 Model, S1 and S2 Tables).

The most common histopathological finding in livers affected by AA-amyloidosis, based on hematoxylin and eosin, was the deposition of abundant eosinophilic material between hepatocytes and sinusoids, causing, when abundant, atrophy and displacement of hepatocytes with derangement of the hepatic cords, which were no longer recognizable (Fig 2a). In some cases, there was also a peri-vascular deposition. In the affected spleens, interstitial and peri-vascular eosinophilic deposits were found (Fig 2b), often resulting in thickening of the trabeculae. In the affected kidneys, amorphous eosinophilic material in the interstitium and/or glomeruli (Fig 2c), multifocal thickening of Bowman's capsule, atrophy or loss of nephrons, fibrosis, and chronic interstitial nephritis were often noted.

## Infectious diseases in cats and SAA fragments in bile

RT-qPCR analysis of 65 spleen samples from 44 cats with and 21 without AA-amyloidosis were tested for infection with feline immunodeficiency virus (FIV), feline leukemia virus (FeLV), and feline coronavirus (FCoV): three (4.6%), 20 (30.8%) and two (3.1%) cases, respectively, were positive. All FIV (100%), 14 FeLV (70.0%), and one FCoV (50.0%) positive cats had AA-amyloidosis in at least one organ (S1 Table). However, statistical analysis did not reveal any association between any of the three infectious diseases and presence of AA-amyloidosis.

Western blot analysis of bile samples from 24 cats revealed that 10 samples from cats with AA-amyloidosis and 5 samples from cats without AA-amyloidosis were SAA-positive, respectively (Fig 3, S4 Fig; negative control for secondary antibody in S5 Fig) showing a band with a molecular mass of approximately 8 kDa. Two of the 10 amyloid-positive cats with AA-amyloidosis did not have any deposits in the liver. Among the SAA-positive bile samples, the quantity of SAA protein in seven, two, and six samples was rated as mild, moderate, and severe, respectively, based on their intensity levels in western blots. Eight amyloid-positive but none of the amyloid-negative cats excreted bile that was rated moderate or severe (p = 0.001). Congo red staining of bile pellets did not detect amyloid fibrils in bile samples, likely due to the low sensitivity of this test; SAA fragments were confirmed with mass spectrometry.

## Variables associated with AA-amyloidosis

Age, gender, and shelter were not associated with AA-amyloidosis nor with the AA-amyloidosis additive score. Duration of stay was not associated with AA-amyloidosis when considered as a dichotomous variable, but was positively associated with the AA-amyloidosis additive score (B: 0.026, CI95%: 0.007–0.046; p = 0.010) (S1 and S2 Models).

In the subset of cats in which the disease duration index was calculated (n = 20, all belonging to shelter C), i.e., the proportion of time a cat was sick due to any other disease during its

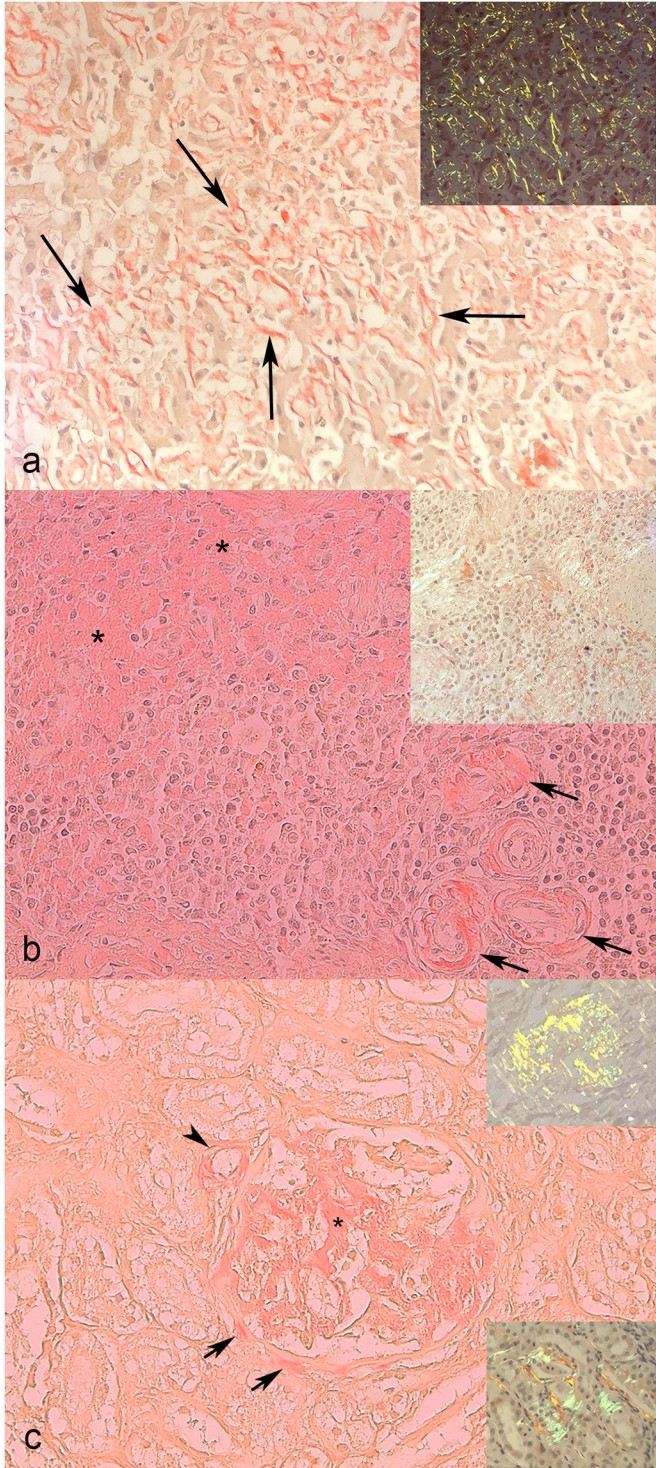

**Fig 2. AA-amylodosis, cat.** Congo red stain. a) Liver. A diffuse moderate amount of red stained amyloid (arrows) displaces the hepatocellular cords, sometimes isolating the hepatocytes. Inset: apple-green birefringence under polarized light. b) Spleen. Red stained amorphous material (amyloid, asterisks) is observed in the interstitium and around arteries (arrows). Inset: apple-green birefringence under polarized light. c) Kidney. The glomerular tuft is expanded by a large amount of Congo red-stained amorphous material (asterisk) consistent with amyloid, which is also present around a proximal arteriole (arrowhead) and along the Bowman's capsule (arrows). Insets: Apple-green birefringence under polarized light is evident in a glomerulus (upper inset) and around tubules (lower inset). Histologic pictures of AA-amyloidosis-negative liver, spleen, and kidney is reported in S3 Fig.

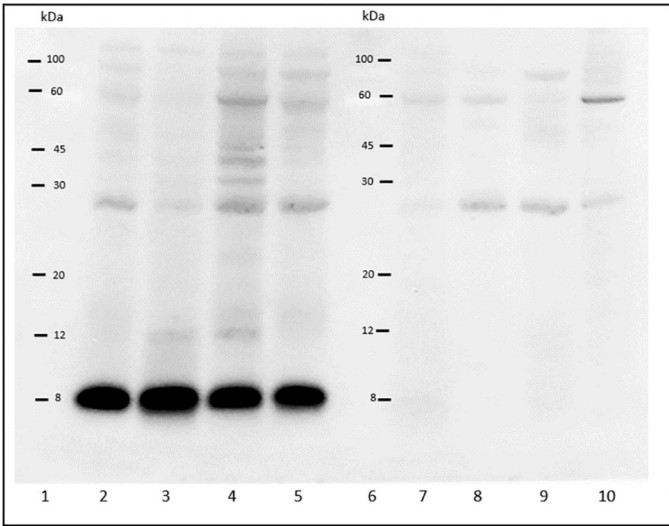

**Fig 3. Representative western blot analysis of bile samples with anti-SAA1 polyclonal antibody.** Lane 1 and 6: pre-stained molecular weight markers; lane 2–5: positive samples; lane 7–10: negative samples.

stay in the shelter, those with AA-amyloidosis showed a higher median value compared to those without [0.129 (Q1-Q3: 0.098–0.233) vs. 0.058 (Q1-Q3: 0.006–0.075); p = 0.020] (Fig 4).

## Discussion

AA-amyloidosis is the most commonly reported amyloid-related disease in many wild and domestic animals such as mouse, cheetah, and cattle [3]. Although large-scale studies on AA-amyloidosis in cats have not been performed, the disease has been described for Abyssinian and Siamese breeds [3, 8]. Few other studies suggested that AA-amyloidosis is a rare disease in

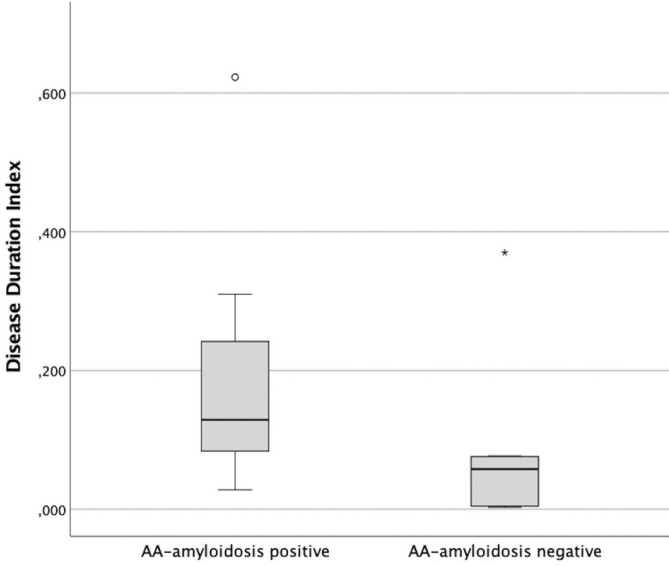

**Fig 4. Box and whisker plot of Disease Duration Index in cats with and without AA-amyloidosis (n = 20, all belonging to shelter C).**

domestic short-hair cats [10, 12, 13]. Indeed, we previously reported only a single amyloid case among 68 client-owned cats that underwent kidney biopsies [25]; Rayhel et al. reported two cases of amyloidosis among 58 cats with renal proteinuria that underwent kidney biopsies [26], confirming AA-amyloidosis as a marginal disease in short-hair cats.

The present investigation revealed a very high prevalence of AA-amyloidosis in dead cats from three shelters in Northern Italy, with 61% of domestic shorthair cats affected. The cat shelters had no direct contact with each other, had an approximate feline population of 200, 60, and 30 cats respectively, and were characterized by high turnover rate and promiscuity. The AA-amyloid deposits were identified in the liver, spleen and kidney by polarized microscopy after Congo red staining and immunofluorescence using SAA-specific antibodies. In a parallel study, the structure of amyloid fibrils extracted from the renal tissue of one of the cats from the present study that was deceased from renal failure, revealed a cross-β arrangement of wild type feline SAA [22].

The frequency of AA-amyloidosis was similar in the liver, spleen, and kidney, with more than 70% of cases displaying concurrent deposits in the three organs. Despite the comparable rate, in affected cats deposits were more severe in the spleen compared to the liver and kidney. It is worth mentioning that in mice the spleen is the organ in which amyloid starts the deposition. Macrophages appear to play a key role in the pathogenesis of amyloidosis in the mice spleen [27]. Whether splenic macrophages also contribute to amyloid deposition in the spleen of cats is currently unknown.

The cause of death in most cats with AA-amyloidosis was renal failure associated with amyloid deposits observed in different parts of the kidney. The post-mortem prevalence of AA-amyloidosis was not significantly different among the three independent shelters, and the duration of the stay in shelters was positively associated with the amount of amyloid deposits in affected cats, suggesting a similar constellation of risk factors. Therefore, risk factors for high prevalence of AA-amyloidosis in shelter-cats appears to be chronic inflammatory diseases, infections, unknown environmental factors, or combinations thereof. In the literature, the deposition of SAA occurred as a consequence of proinflammatory cytokines released during inflammatory or neoplastic disorders [4–6]. The vast majority of our cats were affected by concurrent chronic inflammatory conditions, such as chronic enteropathy and gingivostomatitis, which might have triggered amyloid deposition. In addition, the higher disease duration index observed in cats affected by AA-amyloidosis showed that the longer an animal was sick for any reason, the higher the chance of amyloid deposition. Moreover, overcrowding, which is frequent in shelters, might have facilitated the spread of infectious diseases, such as upper respiratory tract diseases, which are common causes of chronic inflammation in shelter cats. The association between AA-amyloidosis with viral diseases, such as natural FIV infection, has been suggested in previous studies [19, 28]. However, amyloid deposition was not reported in experimentally FIV-infected cats kept in isolation and without concurrent diseases [19], thus suggesting that the lentiviral infection *per se* is not sufficient for the development of AA-amyloidosis. In this study, FIV, FCoV, and FeLV were not associated with the presence of AA-amyloidosis. In our study, only three cats had FIV, and thus it is possible that by including more cats with FIV an association would have been observed. However, the above considerations may not be sufficient to explain the much higher prevalence observed in shelter cats compared to client-owned cats. In the literature, several pathologic conditions have been associated with increased circulating SAA in cats [9] but AA-amyloidosis has been only sporadically reported so far; persistently increased SAA may therefore not be sufficient to initiate amyloid deposition in cats.

Systemic AA-amyloidosis is a well-established case of prion-like disease outside the brain. The transmissibility of AA-amyloidosis has been demonstrated in some animal species

through a prion-like mechanism [17, 29–31]. Fecal shedding of AA-amyloid fibrils contributes to disease onset in captive cheetahs [7], and the incidence of AA-amyloidosis is increased by oral ingestion of amyloid fibrils in mice [1]. Orofecal transmission has thus been proposed to play an important role in the development of AA-amyloidosis in captive animals [7, 17]. Horizontal transmission might be relevant to the disease in shelter cats, but remains to be confirmed. The molecular weight of SAA fragments found in bile samples (Fig 3) is compatible with SAA fragments assembled in SAA fibrils extracted from the renal tissue of an affected shelter cat: such fragments comprise 75 residues (from residue 19 to 94) [22]. Although the presence of SAA fragments in bile samples has been shown herein, its folding, as native or aggregated, could not be established. The identification of SAA fragments in bile of cats not affected by amyloidosis may indicate that its excretion is not a consequence of the disease. Whether SAA fragments in bile play a role in the pathogenesis of amyloidosis, possibly undergoing conformational changes into fibrils in faeces, needs to be clarified.

The study has some limitations. Firstly, the majority of the medical records were incomplete. In several cases it was thus not possible to determine the clinical relevance of amyloid deposition in most cases. It is probable that some of the cats with AA-amyloidosis were asymptomatic for the disease.

In conclusion, this study demonstrates that the post-mortem prevalence of AA-amyloidosis is high in domestic short-hair cats housed in shelters. This observation is in contrast with observations in client-owned cats [25, 26]. Whether bile excretion of AA-fibrils represents a vehicle of transmission explaining the high frequency of the disease in shelter cats remains to be elucidated. The time spent in shelters is associated with the amount of AA-amyloid deposits; in particular, the longer an animal was sick for any reason, the higher the chance of amyloid deposition, Intriguingly, cat shelters may represent a model to investigate the spread of AA-amyloidosis in shelters and farms animals as well as in captive wild animals.

## Supporting information

**S1 Fig. Fluorescence-immunohistochemistry of cat pancreatic tissue negative for AA-amyloidosis.** Nuclei are blue with DAPI (a), no amyloid is identified with Thioflavine S (b) or anti-SAA (c).
(TIFF)

**S2 Fig. Fluorescence-immunohistochemistry of cat kidney tissue positive for AA-amyloidosis.** Nuclei are blue with DAPI(a,d; in blue), amyloid is identified with Thioflavine S (b,e; in green) and anti-SAA (c, in red). No signal was detected when the secondary antibody Alexa FluorTM546 was used without anti-SAA (f).
(TIFF)

**S3 Fig. Cat, negative controls from liver (a), spleen (b), and kidney (c).** No interstitial red material is detected by Congo red stain.
(TIF)

**S4 Fig. Western blot analysis of bile samples with anti-SAA1 polyclonal antibody.** The intensity levels in positive western blot are marked as negative (-), mild (+), moderate (++) and severe (+++). Lanes 5, 11, 13, 17, 25: molecular weight markers; lanes 1–3, 6–10, 16, 18, 20, 21–24: positive bile samples. Lanes 4, 12, 14, 15, 19, 26–29: negative bile samples.
(TIF)

**S5 Fig. Western blot analysis of a bile sample with amyloid fragments in a cat.** Panel A, with secondary antibody alone. Panel B, with both primary and secondary antibodies. Lane 1:

pre-stained molecular markers; lane 2: bile sample with amyloid fragments.
(TIF)

**S1 Model. Logistic regression model for AA-amyloidosis status (presence vs absence).**
(DOCX)

**S2 Model. Generalized linear model for AA-amyloidosis additive score.**
(DOCX)

**S1 Table. Data of all the included cats: N˚ of cat, shelter, gender, age at death (years), duration of stay in the shelter (months), presence of amyloidosis in any of the three investigated organs, presence of hepatic amyloidosis, presence of splenic amyloidosis, presence of renal amyloidosis, histological amyloid liver score, histological amyloid splenic score, histological amyloid kidney score, amyloidosis score, results of PCR for FIV, results of PCR for FeLV, results of PCR for FCoV, other concurrent chronic diseases, cause of death, western blot result.**
(DOCX)

**S2 Table. Distribution of the AA-amyloidosis additive score in cats.**
(DOCX)

## Acknowledgments

Valter Fiore (Associazione La Cincia, Val Della Torre, Torino, Italy), Daniela Monfroglio and Davide Pozzi (Associazione Amici dei Gatti Onlus, Galliate, Novara, Italy), Eraldo Bellini (Parco Animalista Piazza D'Armi Onlus, Torino, Italy), Angela Barbato, Federica Folatti, and Francesca Iavazzo are kindly acknowledged for their support.

## Author Contributions

**Conceptualization:** Filippo Ferri, Silvia Ferro, Federico Porporato, Enrico Gallo, Luigi Michele Coppola, Gabriele Gerardi, Giampaolo Merlini, Eric Zini.

**Data curation:** Filippo Ferri, Silvia Ferro, Chiara Guglielmetti, Maria Mazza, Marta Ferrero, Michele Drigo, Monique Vogel, Federico Storni, Martin F. Bachmann, Anne-Cathrine Vogt, Serena Caminito, Giulia Mazzini, Francesca Lavatelli, Giovanni Palladini.

**Formal analysis:** Filippo Ferri, Silvia Ferro, Carolina Callegari, Maria Mazza, Marta Ferrero, Chiara Crinò, Michele Drigo, Gabriele Gerardi, Anne-Cathrine Vogt, Giulia Mazzini, Francesca Lavatelli, Giovanni Palladini, Giampaolo Merlini.

**Funding acquisition:** Maria Mazza, Stefano Ricagno, Eric Zini.

**Investigation:** Filippo Ferri, Silvia Ferro, Chiara Guglielmetti, Maria Mazza, Marta Ferrero, Monique Vogel, Federico Storni, Martin F. Bachmann, Anne-Cathrine Vogt, Serena Caminito, Giulia Mazzini, Francesca Lavatelli.

**Methodology:** Enrico Gallo, Michele Drigo, Anne-Cathrine Vogt, Serena Caminito, Giovanni Palladini, Giampaolo Merlini, Eric Zini.

**Project administration:** Gabriele Gerardi, Eric Zini.

**Supervision:** Eric Zini.

**Writing – original draft:** Filippo Ferri, Federico Porporato, Chiara Crinò, Eric Zini.

**Writing – review & editing:** Filippo Ferri, Silvia Ferro, Federico Porporato, Carolina Callegari, Tim Paul Schulte, Stefano Ricagno, Eric Zini.

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
