## [Decision Letter · Decision Letter 0]

29 Nov 2022

PONE-D-22-29069AA-amyloidosis in cats (Felis catus) housed in sheltersPLOS ONE

Dear Dr. Ferri,

Thank you for submitting your manuscript to PLOS ONE. After careful consideration, we feel that it has merit but does not fully meet PLOS ONE’s publication criteria as it currently stands. Therefore, we invite you to submit a revised version of the manuscript that addresses the points raised during the review process.

We look forward to receiving your revised manuscript.

Kind regards,

Maria Stefania Latrofa

Academic Editor

PLOS ONE

2. To comply with PLOS ONE submissions requirements, in your Methods section, please provide additional information on the animal research (mice) and ensure you have included details on (1) methods of sacrifice, (2) methods of anesthesia and/or analgesia, and (3) efforts to alleviate suffering.

3. Thank you for stating the following in the Grant or support Section of your manuscript: 

"BIRD 193233 University of Padova; 10/19 RC Istituto Zooprofilattico Sperimentale del Piemonte Liguria e Valle d'Aosta; 2019 and 2020 Research Funds AniCura. This study was partially supported by Ricerca Corrente funding from Italian Ministry of Health to IRCCS Policlinico San Donato."

"Eric Zini: BIRD 193233 University of Padova;  2019 and 2020 Research Funds AniCura.

Maria Mazza: 10/19 RC Istituto Zooprofilattico Sperimentale del Piemonte Liguria e Valle d'Aosta. 

Stefano Ricagno: this study was partially supported by Ricerca Corrente funding from Italian Ministry of Health to IRCCS Policlinico San Donato.

Reviewers' comments:

Reviewer's Responses to Questions

**Comments to the Author**

1. Is the manuscript technically sound, and do the data support the conclusions?

Reviewer #1: Partly

Reviewer #2: Partly

2. Has the statistical analysis been performed appropriately and rigorously? 

Reviewer #1: Yes

Reviewer #2: I Don't Know

3. Have the authors made all data underlying the findings in their manuscript fully available?

Reviewer #1: No

Reviewer #2: No

4. Is the manuscript presented in an intelligible fashion and written in standard English?

Reviewer #1: Yes

Reviewer #2: No

5. Review Comments to the Author

Reviewer #1: Dear Authors,

please find below my comments.

Lines 130-131: why is Congo Red used? It is necessary to specify that "staining with Congo Red (CR) is a qualitative method used for the identification of amyloids in vitro and in tissue sections" and to correctly quote the bibliography.

Lines 165-166: "Multiple techniques (Congo red staining, western blotting)". Being only two methods, I would eliminate "multiple techiques".

Lines 209-2011: it is not clear how the "disease duration index" was calculated. It is reported that "disease duration index was calculated arbitrarily" and that "The index represented the proportion of time a cat was sick during its stay in the shelter". Is the index therefore the mathematical ratio? Why is it written "arbitrarily"?

Line 263: "A co-localization staining" between what (Thioflavine S, DAPI or IgG)? Please specify better.

Lines 126-147: negative controls were not used in the immunofluorescence experiments and this is not a correct methodological approach for this method. The experiment requires negative controls.

Fig 2: the images in the file that I have downloaded are of low quality, in some points pixelated and this does not allow to evaluate the results obtained.

Fig 2: to understand the significance of the results, negative controls showing the histology of non-affected individuals are required.

Lines 314-323: I personally believe that a scientific work must show all the results obtained also to verify the truthfulness of the results and the correctness of the interpretations and conclusions. The number of samples examined by western blot is high (bile sample from 24) but not so high to make it impossible to present all the results, both positive and negative. I believe that the results of all western blots should be presented and not only those of some representative results, as shown in Fig. 3. Please report all results obtained (if necessary use supplementary materials).

Figure 3: "pre-stained molecular weight markers" are used in the experiment but are not visible in the result. Molecular weight numbers lack a dash "-" that indicates exactly the height that the molecular weight refers to so that the molecular weight of the bands can be accurately assessed.

Figure 3: What exactly is the molecular weight of the positive bands? I do not think it is ever made explicit in the text.

Figure 3: the experiment lacks the negative control consisting of the use of only the secondary antibody.

Lines 405-407 the molecular weight of the bands is not explained.

Lines 419-420: "This observation is in contrast with observations in client-owned cats". Where does this observation come from? Are you referring to data reported in this study or to results of other not cited works?

Reviewer #2: The manuscript “AA-amyloidosis in cats (Felis catus) housed in shelters” describes the detection in a high percentage of amyloid fibrils in tissues and bile of cats that died in shelters for different reasons. The authors conclude that the long staying in the shelter and co-presence of diseases increase the chance of amyloid deposition and add some speculations.

The topic of cat amyloidosis is interesting also for several human diseases, and the manuscript data might be of general interest. However, It has been difficult and frustrating to follow the sampling and result description as it was confounded which animal had which phenotype or was analyzed. Moreover without a plain clear and transparent display of all the raw data, the point of all the work is only what I just synthetized and what is even more critical the individual results are of little use as they are presented. Material and Methods together with the Results section must be improved to obtain a clearer exposition and results convincing. So, I suggest to the authors to add all the raw data in a Supplementary material Table, to allow a clear understanding of their analyses and conclusions.

In my opinion, to be re-considered for acceptance the paper must address this request of raw data disclosure, together with the comments that follow.

INTRODUCTION

Lines 69 and 72. As your reference (3) is about cats, please anticipate “In cats” to line 69 and remove it from line 74 as follows: “In cats the exact pathogenesis…[3]. Many conditions…”.

Line 74. “but only a limited number of cats eventually develop the disease [9]”.

Assuming “the disease” refers to amyloidosis, the reference [9] is misplaced and generates misleading information. It should be moved after “high production of SAA”, in case.

Thus, the sentence in line 74 needs a reference in support.

Tamamoto et al 2013 do not have amyloidosis among their considered diagnostic categories, nor report analyses made to detect the presence/absence of amyloidosis in their samples. They mention amyloidosis only in their discussion as follows: “SAA protein is known as a precursor of amyloid A fibrils, and sustained high SAA levels may give rise to secondary amyloidosis, so-called reactive amyloid A amyloidosis. (8) The development of amyloid A amyloidosis is considered an important prognostic factor. (5)…..However, all of these facts were reported in human medicine, and it remains unclear whether it can be applied to cats.”

Line 76. The author might want to complete the sentence: “the differences in SAA genetic sequence proved to not contribute to the pathogenesis of feline AA-amyloidosis both in Japanese domestic cats [10] and in Abyssinian breed [doi: 10.1038/s41598-021-87168-0.]

Line 81-83. Please detail more specifically (see underlined words): “the development of AA-amyloidosis in mice induced by injection of an inflammatory stimulant appears to be accelerated by the oral administration of semipurified amyloid fibrils extracted from murine, humans and cattle affected tissues and lead to the hypothesis of both intra-species and cross-species transmissibility [13].

Please add “intra species” because the focus is on the cat, the cross-species is interesting, but here less important.

Line 96. “3 consecutive”. Please change “three”, and what do you mean by “consecutive”? In a short period?

Line 97. Please correct and add: “within six months from the admission”

MATERIAL AND METHODS

Line 104 and line 229 (Results). Please add to the main text in MM the number of cats analyzed (79).

Moreover, in Results, the authors report 80 collected cats, but then you exclude one because of late sampling, which is the only parameter of exclusion stated in MM. So, no reason to talk of 80 cats, here again please report 79.

Please specify in MM that necropsy report/clinical anamnesis were available for n …(?)…/79 cats

In addition, it might be of interest which approximate percentage of each shelter total population were those cats and some general description of shelter structure and management if they might have or not have an impact on the cat promiscuity and epidemiology.

Lines 104, 231, and 354. “Northern Italy”

Line 202. Please, do you mean “both for each shelter and the entire population of sampled cats”?

RESULTS

Overall it is difficult to follow the data throughout the results section. The authors are warmly invited to produce a synoptic table in the supplementary material with all the raw data of each cat, even with missing data (see below)

Lines 238-244. This paragraph is confounding.

In Table S1 how many cats showed the 47 causes of death reported? It should be assumed 47 cats. In Table S1 are reported, “other diseases” (36 diseases) on 22 cats from shelter C. Why these data are not merged in a single table? Are the cats with phenotype 47+22=69 or what?

Please remove S1T and S2T and list in one Supplementary single table all the 79 cats, their individual phenotypic data collected when available (disease and cause of death), positiveness to Congo (negativeness and additional score or at least the presence/absence of amyloid) and the shelter.

The authors are also strongly encouraged to integrate this new Table with cat age when accepted in the shelter (if available), time of permanence in the shelter, age at death

Table 1. please specify in table 1 legend the number of cats such as “Table 1. Gender distribution, age and length of stay of the 79 cats in the three shelters”.

Line 267- 273. “Of the 22 cats with available information on other diseases”, what about the cats listed in S1T (n.47 )? Were the remaining without phenotype, which ones (see above)?

Line 272-274 “Of the 22 cats with available information on other (36?) diseases, 12 showed AA-amyloidosis”. Does this Mean that 10/22 = 45,6% of those affected by other diseases cats did not develop amyloidosis? Please clarify and in case comment.

“…with 11 cats (91.7%) being affected by one or more concurrent diseases.” Meaning that in a subset of 12, out of a total of 22 cats all with phenotype, only 91% had phenotype? Please clarify.

Table 2. Please specify also in Table 2 legend “in 79 cats”

Line 307. Please specify which diseases were the 65 cats affected by, in case they were (in addition to FILV, FeLV, and FCoV). It should be perfect in the New Supplementary table reporting the phenotypes to add a column where the cats analyzed are ticked

Line 314. Please specify which diseases or viral infections were the 24 cats affected by, apart from amyloidosis. It should be perfect in the New S. table reporting the phenotypes to add another column where the cats analyzed are ticked

Lines 335 – 338. Please specify which subset of cats (numerosity, shelter, etc). Please specify it also in Fig. 4

DISCUSSION

Line 417. “It is probable that some of the cats with AA amyloidosis were asymptomatic for the disease”

In domestic cat amyloidosis is essentially “asymptomatic” until almost the end, and the late symptoms are not pathognomonic. The diagnosis is basically by biopsy or necropsy and Congo red stain. If deposits in tissues were found the cat was affected by amyloidosis, if not the cat was not affected. Then please clarify the rationale of this sentence.

Line 420. “Whether bile excretion of AA-fibrils represents a vehicle of transmission explaining the high frequency of the disease in shelter cats remains to be elucidated.” Did the authors collect gastro-intestinal mucosa and feces, having planned to analyze bile in a prospective study? In case this may be specified as further investigation.

6. PLOS authors have the option to publish the peer review history of their article (what does this mean?). If published, this will include your full peer review and any attached files.

Reviewer #1: No

Reviewer #2: No

---

## [Author Response · Author response to Decision Letter 0]

2 Jan 2023

Dear Editor and Reviewers,

The authors thank the editor and the reviewers for their comments; you can please find below the answers.

Editor’s comment

AUTHORS: the issue has been addressed and the manuscript modified.

2. To comply with PLOS ONE submissions requirements, in your Methods section, please provide additional information on the animal research (mice) and ensure you have included details on (1) methods of sacrifice, (2) methods of anesthesia and/or analgesia, and (3) efforts to alleviate suffering.

AUTHORS: the issue has been addressed and the manuscript modified.

3. Thank you for stating the following in the Grant or support Section of your manuscript: 

"BIRD 193233 University of Padova; 10/19 RC Istituto Zooprofilattico Sperimentale del Piemonte Liguria e Valle d'Aosta; 2019 and 2020 Research Funds AniCura. This study was partially supported by Ricerca Corrente funding from Italian Ministry of Health to IRCCS Policlinico San Donato."

"Eric Zini: BIRD 193233 University of Padova; 2019 and 2020 Research Funds AniCura.

Maria Mazza: 10/19 RC Istituto Zooprofilattico Sperimentale del Piemonte Liguria e Valle d'Aosta. 

Stefano Ricagno: this study was partially supported by Ricerca Corrente funding from Italian Ministry of Health to IRCCS Policlinico San Donato.

AUTHORS: the funding information has been removed from the manuscript. The funding information and the Funding Statement in the online submission form are reporting the same information but simply in a different order.

AUTHORS: from my account the ORCID iD seems to be regularly present in “my information” page. My ORCID iD number is 0000-0002-6727-2399.

AUTHORS: the ethics statements have been removed in all other sections besides the Methods.

Reviewer #1: Dear Authors,

please find below my comments.

Lines 130-131: why is Congo Red used? It is necessary to specify that "staining with Congo Red (CR) is a qualitative method used for the identification of amyloids in vitro and in tissue sections" and to correctly quote the bibliography.

AUTHORS: the issue has been addressed and the manuscript modified.

Lines 165-166: "Multiple techniques (Congo red staining, western blotting)". Being only two methods, I would eliminate "multiple techiques".

AUTHORS: the issue has been addressed and the manuscript modified.

Lines 209-2011: it is not clear how the "disease duration index" was calculated. It is reported that "disease duration index was calculated arbitrarily" and that "The index represented the proportion of time a cat was sick during its stay in the shelter". Is the index therefore the mathematical ratio? Why is it written "arbitrarily"?

AUTHORS: The index is the mathematical ratio. The word “arbitrarily” has been removed from the manuscript.

Line 263: "A co-localization staining" between what (Thioflavine S, DAPI or IgG)? Please specify better.

AUTHORS: Amyloidogenic aggregates were stained with ThioflavineS dye and sera from vaccinated mice. A merge image was created to visualize the co-localization of ThioS and IgG. The manuscript has been modified.

Lines 126-147: negative controls were not used in the immunofluorescence experiments and this is not a correct methodological approach for this method. The experiment requires negative controls.

AUTHORS: This issue has been addressed. Negative controls had been performed but were not mentioned in the manuscript. The manuscript has been modified and figures added as supplementary materials. 

Fig 2: the images in the file that I have downloaded are of low quality, in some points pixelated and this does not allow to evaluate the results obtained.

AUTHORS: the figures were prepared according to PLOS ONE figure guidelines. The authors have repeated the submission if technical issues previously occurred. The reviewer should find a link to the high quality figures in the corner of the proof.

Fig 2: to understand the significance of the results, negative controls showing the histology of non-affected individuals are required.

AUTHORS: Negative controls of non-affected cats have been added as supplementary materials.

Lines 314-323: I personally believe that a scientific work must show all the results obtained also to verify the truthfulness of the results and the correctness of the interpretations and conclusions. The number of samples examined by western blot is high (bile sample from 24) but not so high to make it impossible to present all the results, both positive and negative. I believe that the results of all western blots should be presented and not only those of some representative results, as shown in Fig. 3. Please report all results obtained (if necessary use supplementary materials).

AUTHORS: A Supplementary figure with all the results of the western blot has been added; the authors recognize the low quality of the picture and would prefer not to publish it.

Figure 3: "pre-stained molecular weight markers" are used in the experiment but are not visible in the result. Molecular weight numbers lack a dash "-" that indicates exactly the height that the molecular weight refers to so that the molecular weight of the bands can be accurately assessed.

AUTHORS: In the supplementary image (S4 Fig.), molecular weights are visible with chemoluminescence. In Figure 3 dashes have been inserted at each molecular weight band.

Figure 3: What exactly is the molecular weight of the positive bands? I do not think it is ever made explicit in the text.

AUTHORS: The molecular weight of the positive bands is 8 kDa. The manuscript and Fig 3 have been modified accordingly.

Figure 3: the experiment lacks the negative control consisting of the use of only the secondary antibody.

AUTHORS: A negative control test was performed and pictures added as supplementary materials (S5 Fig.). The supplementary figure S2 shows the analysis of a bile sample with amyloid fragments with both primary and secondary antibodies as well as with secondary antibody alone. The absence of signal with the secondary antibody alone reveals the specificity of the 8 kDa band.

Lines 405-407 the molecular weight of the bands is not explained.

AUTHORS: The molecular weight of SAA fragments found in bile samples is compatible with SAA fragments assembled in SAA fibrils extracted from the renal tissue of an affected shelter cat: such fragments comprise 75 residues (from residue 19 to 94). The manuscript has been modified.

Lines 419-420: "This observation is in contrast with observations in client-owned cats". Where does this observation come from? Are you referring to data reported in this study or to results of other not cited works?

AUTHORS: two references have been added. In particular, Rossi et al. reported only one case of amyloidosis among 68 client-owned cats that underwent renal biopsy and Rayhel et al. reported two cases of amyloidosis among 58 cats with renal proteinuria.

Reviewer #2: The manuscript “AA-amyloidosis in cats (Felis catus) housed in shelters” describes the detection in a high percentage of amyloid fibrils in tissues and bile of cats that died in shelters for different reasons. The authors conclude that the long staying in the shelter and co-presence of diseases increase the chance of amyloid deposition and add some speculations.

The topic of cat amyloidosis is interesting also for several human diseases, and the manuscript data might be of general interest. However, It has been difficult and frustrating to follow the sampling and result description as it was confounded which animal had which phenotype or was analyzed. Moreover without a plain clear and transparent display of all the raw data, the point of all the work is only what I just synthetized and what is even more critical the individual results are of little use as they are presented. Material and Methods together with the Results section must be improved to obtain a clearer exposition and results convincing. So, I suggest to the authors to add all the raw data in a Supplementary material Table, to allow a clear understanding of their analyses and conclusions.

In my opinion, to be re-considered for acceptance the paper must address this request of raw data disclosure, together with the comments that follow.

AUTHORS: A table with the raw data has been prepared and added as supplementary material; the table includes the cat, shelter, gender, age at death (years), duration of stay in the shelter (months), presence of amyloidosis in any of the three investigated organs, presence of hepatic amyloidosis, presence of splenic amyloidosis, presence of renal amyloidosis, histological amyloid liver score, histological amyloid splenic score, histological amyloid kidney score, amyloidosis score, result of PCR for FIV, result of PCR for FeLV, result of PCR for FCoV, other concurrent chronic diseases, cause of death, western blot result.

INTRODUCTION

Lines 69 and 72. As your reference (3) is about cats, please anticipate “In cats” to line 69 and remove it from line 74 as follows: “In cats the exact pathogenesis…[3]. Many conditions…”.

AUTHORS: the issue has been addressed and the manuscript modified.

Line 74. “but only a limited number of cats eventually develop the disease [9]”.

Assuming “the disease” refers to amyloidosis, the reference [9] is misplaced and generates misleading information. It should be moved after “high production of SAA”, in case.

Thus, the sentence in line 74 needs a reference in support.

Tamamoto et al 2013 do not have amyloidosis among their considered diagnostic categories, nor report analyses made to detect the presence/absence of amyloidosis in their samples. They mention amyloidosis only in their discussion as follows: “SAA protein is known as a precursor of amyloid A fibrils, and sustained high SAA levels may give rise to secondary amyloidosis, so-called reactive amyloid A amyloidosis. (8) The development of amyloid A amyloidosis is considered an important prognostic factor. (5)…..However, all of these facts were reported in human medicine, and it remains unclear whether it can be applied to cats.”

AUTHORS: The authors deem interesting that many disorders are associated with high circulating concentrations of SAA in cats (as described by Tamamoto et al 2013) despite a limited number has been so far reported with amyloidosis in the literature. The authors are therefore speculating that high concentrations of SAA might not be sufficient to promote amyloidosis in cats. The reference has been replaced and the manuscript modified.

Line 76. The author might want to complete the sentence: “the differences in SAA genetic sequence proved to not contribute to the pathogenesis of feline AA-amyloidosis both in Japanese domestic cats [10] and in Abyssinian breed [doi: 10.1038/s41598-021-87168-0.]

AUTHORS: the issue has been addressed and the manuscript modified.

Line 81-83. Please detail more specifically (see underlined words): “the development of AA-amyloidosis in mice induced by injection of an inflammatory stimulant appears to be accelerated by the oral administration of semipurified amyloid fibrils extracted from murine, humans and cattle affected tissues and lead to the hypothesis of both intra-species and cross-species transmissibility [13].

Please add “intra species” because the focus is on the cat, the cross-species is interesting, but here less important.

AUTHORS: the issue has been addressed and the manuscript modified.

Line 96. “3 consecutive”. Please change “three”, and what do you mean by “consecutive”? In a short period?

AUTHORS: The manuscript has been modified accordingly; “consecutive” has been removed. The authors have previously written “consecutive” because amyloidosis has been diagnosed in three cats from the same shelter over a period of 6 months, making its occurrence particularly noticeable.

Line 97. Please correct and add: “within six months from the admission”

AUTHORS: As above mentioned, the authors have decided to investigate amyloidosis in shelter cats because 3 different cats were diagnosed with the disease over a period of 6 months. Therefore “within 6 months” does not refer to the time spanning from admission to diagnosis of amyloidosis in each cat, but to the overall period in which the 3 cats were diagnosed with the protein misfolding disease. 

The authors have modified “within 6 months” into “within six months”.

MATERIAL AND METHODS

Line 104 and line 229 (Results). Please add to the main text in MM the number of cats analyzed (79).

AUTHORS: the issue has been addressed and the manuscript modified.

Moreover, in Results, the authors report 80 collected cats, but then you exclude one because of late sampling, which is the only parameter of exclusion stated in MM. So, no reason to talk of 80 cats, here again please report 79.

AUTHORS: The manuscript has been modified.

Please specify in MM that necropsy report/clinical anamnesis were available for n …(?)…/79 cats

AUTHORS: The manuscript has been modified. The shelters are located approximately 100 km one from the other; in addition, in order to be able to have histologic samples of good quality, the authors have decided to collect specimens within six hours from death. The technical staff was instructed on how to adequately collect samples but were unable to perform full necropsies; hence, reports of the full necropsy are not available.

In addition, it might be of interest which approximate percentage of each shelter total population were those cats and some general description of shelter structure and management if they might have or not have an impact on the cat promiscuity and epidemiology.

AUTHORS: the three cat shelters had an approximate population of 200, 60, and 30 cats respectively, all with high turnover rate and promiscuity; a comment has been added in the discussion of the manuscript to acknowledge this observation. The number of deaths is calculated over a 9-months period (January-September 2019); the high turnover in each shelter was caused by the interplay between deaths, adoptions and new admissions.

Lines 104, 231, and 354. “Northern Italy”

AUTHORS: the issue has been addressed and the manuscript modified.

Line 202. Please, do you mean “both for each shelter and the entire population of sampled cats”?

AUTHORS: The frequencies were calculated on the population of sampled cats. The manuscript has been modified.

RESULTS

Overall it is difficult to follow the data throughout the results section. The authors are warmly invited to produce a synoptic table in the supplementary material with all the raw data of each cat, even with missing data (see below)

AUTHORS: A table with the raw data has been prepared and added as supplementary material; the table includes the n° of cat, shelter, gender, age at death (years), duration of stay in the shelter (months), presence of amyloidosis in any of the three investigated organs, presence of hepatic amyloidosis, presence of splenic amyloidosis, presence of renal amyloidosis, histological amyloid liver score, histological amyloid splenic score, histological amyloid kidney score, amyloidosis score, result of PCR for FIV, result of PCR for FeLV, result of PCR for FCoV, other concurrent chronic diseases, cause of death, western blot result.

Lines 238-244. This paragraph is confounding.

In Table S1 how many cats showed the 47 causes of death reported? It should be assumed 47 cats. In Table S1 are reported, “other diseases” (36 diseases) on 22 cats from shelter C. Why these data are not merged in a single table? Are the cats with phenotype 47+22=69 or what?

Please remove S1T and S2T and list in one Supplementary single table all the 79 cats, their individual phenotypic data collected when available (disease and cause of death), positiveness to Congo (negativeness and additional score or at least the presence/absence of amyloid) and the shelter.

The authors are also strongly encouraged to integrate this new Table with cat age when accepted in the shelter (if available), time of permanence in the shelter, age at death

AUTHORS: the new supplementary table has been created and added; S1T and S2T have been removed.

Table 1. please specify in table 1 legend the number of cats such as “Table 1. Gender distribution, age and length of stay of the 79 cats in the three shelters”.

AUTHORS: the issue has been addressed and the manuscript modified.

Line 267- 273. “Of the 22 cats with available information on other diseases”, what about the cats listed in S1T (n.47 )? Were the remaining without phenotype, which ones (see above)?

AUTHORS: the cats included in the present manuscript were housed in shelters with a high turnover of animals. These facilities are on a very restricted budget, shortage of personnel and have very incomplete databases. As a consequence, it was particularly challenging to gather clinical information from the animals. Of note, cat shelter “C” has a different management due to more resources, less animals, and an acceptable database, which allowed the authors to collect clinical information on concurrent chronic diseases (i.e., overall from 22 cats, some of which with more than one concurrent disease). Concerning the cause of death of all the 79 cats included,12 were euthanized due to worsening of their clinical conditions and 20 were found dead without any obvious reason (and without previous diagnostic workup). Overall, results of investigations were available to determine the cause of death in 47 cats. The authors added a supplementary table with the details.

Line 272-274 “Of the 22 cats with available information on other (36?) diseases, 12 showed AA-amyloidosis”. Does this Mean that 10/22 = 45,6% of those affected by other diseases cats did not develop amyloidosis? Please clarify and in case comment.

“…with 11 cats (91.7%) being affected by one or more concurrent diseases.” Meaning that in a subset of 12, out of a total of 22 cats all with phenotype, only 91% had phenotype? Please clarify.

AUTHORS: The reviewer interpreted correctly; of the 22 cats with available information on concurrent chronic diseases 10 did not develop amyloidosis and 12 developed the amyloidosis. Among the 12 cats with amyloidosis, 11 had concurrent chronic diseases and 1 was not affected by any concurrent chronic disease. A sentence was added in the manuscript.

Table 2. Please specify also in Table 2 legend “in 79 cats”

AUTHORS: the issue has been addressed and the manuscript modified.

Line 307. Please specify which diseases were the 65 cats affected by, in case they were (in addition to FILV, FeLV, and FCoV). It should be perfect in the New Supplementary table reporting the phenotypes to add a column where the cats analyzed are ticked. 

AUTHORS: the issues has been addressed and the supplementary table added to include viral infections, concurrent chronic diseases and cause of death.

Line 314. Please specify which diseases or viral infections were the 24 cats affected by, apart from amyloidosis. It should be perfect in the New S. table reporting the phenotypes to add another column where the cats analyzed are ticked

AUTHORS: the issues has been addressed and the supplementary table added.

Lines 335 – 338. Please specify which subset of cats (numerosity, shelter, etc). Please specify it also in Fig. 4

AUTHORS: the issue has been addressed, the manuscript and the caption of Fig. 4 modified.

DISCUSSION

Line 417. “It is probable that some of the cats with AA amyloidosis were asymptomatic for the disease”

In domestic cat amyloidosis is essentially “asymptomatic” until almost the end, and the late symptoms are not pathognomonic. The diagnosis is basically by biopsy or necropsy and Congo red stain. If deposits in tissues were found the cat was affected by amyloidosis, if not the cat was not affected. Then please clarify the rationale of this sentence.

AUTHORS: AA-amyloidosis has been sporadically reported in non-predisposed breeds and, in those cases, it was associated with clinical signs and/or laboratory abnormalities such as those of hepatic rupture (Zini et al. 2008 – “Transient laboratory changes in a cat with repeated hepatic haemorrhages due to amyloidosis”), pulmonary haemorrhage (Mawby et al. 2018 – “Fatal pulmonary haemorrhage associated with vascular amyloid deposition in a cat”), or proteinuria (Rossi et al. 2019, “Immune-complex glomerulonephritis in cats: a retrospective study based on clinico-pathological data, histopathology and ultrastructural features.”). In addition, Paltrinieri et al (2014, “Changes in serum and urine SAA concentrations and qualitative and quantitative proteinuria in Abyssinian cats with familial amyloidosis: a five-year longitudinal study”) described a high frequency of proteinuria along with signs of illness in Abyssian cats affected by familial amyloidosis; in the discussion the authors reported that those cats often had died within 5 years due to the deposition of amyloid. Due to the above, currently the authors think that the literature does not point to amyloidosis as an “essentially asymptomatic” disease. Nonetheless, we agree with the reviewer comment that most cats are asymptomatic until almost the end. 

Line 420. “Whether bile excretion of AA-fibrils represents a vehicle of transmission explaining the high frequency of the disease in shelter cats remains to be elucidated.” Did the authors collect gastro-intestinal mucosa and feces, having planned to analyze bile in a prospective study? In case this may be specified as further investigation.

AUTHORS: The authors initially planned to collect feces and urine of cats but the task turned out to be extremely difficult because of the high promiscuity in each shelter; in addition, many cats urinated at death, making urine collection impossible in the majority of the cases. Nevertheless, fecal samples had been collected from a small subset of animals; unfortunately, the quality of the western blots was poor due to the intrinsic complexity of the substrate. Of note, the authors are currently planning a study to verify if other organs are involved by amyloid deposition.

Kind regards,

The authors.

---

## [Decision Letter · Decision Letter 1]

1 Feb 2023

AA-amyloidosis in cats (Felis catus) housed in shelters

PONE-D-22-29069R1

Dear Dr. Ferri,

We’re pleased to inform you that your manuscript has been judged scientifically suitable for publication and will be formally accepted for publication once it meets all outstanding technical requirements.

Kind regards,

Maria Stefania Latrofa

Academic Editor

PLOS ONE

Additional Editor Comments:

The manuscript is considered accepted. It requires a minor change as noted by reviewer 2.

Reviewers' comments:

Reviewer's Responses to Questions

**Comments to the Author**

1. If the authors have adequately addressed your comments raised in a previous round of review and you feel that this manuscript is now acceptable for publication, you may indicate that here to bypass the “Comments to the Author” section, enter your conflict of interest statement in the “Confidential to Editor” section, and submit your "Accept" recommendation.

Reviewer #1: All comments have been addressed

Reviewer #2: (No Response)

2. Is the manuscript technically sound, and do the data support the conclusions?

Reviewer #1: Yes

Reviewer #2: Yes

3. Has the statistical analysis been performed appropriately and rigorously? 

Reviewer #1: Yes

Reviewer #2: Yes

4. Have the authors made all data underlying the findings in their manuscript fully available?

Reviewer #1: Yes

Reviewer #2: Yes

5. Is the manuscript presented in an intelligible fashion and written in standard English?

Reviewer #1: Yes

Reviewer #2: Yes

6. Review Comments to the Author

Reviewer #1: (No Response)

Reviewer #2: The authors have adequately addressed my comments raised in the previous round of review.

I have only one remark that should be addressed and was not completely, in my opinion: in my previous review I requested to specify in MM how many cases were necropsied.

The authors kindly replied that "...The technical staff was instructed on how to adequately collect samples but were unable to perform full necropsies; hence, reports of the full necropsy are not available". This is very understandable, but I feel it is important to specify this in the Materials and Methods with a short sentence such as

New Line 112 "...tissue samples had been collected within six hours of death, not allowing for a full necropsy to be recorded"

7. PLOS authors have the option to publish the peer review history of their article (what does this mean?). If published, this will include your full peer review and any attached files.

Reviewer #1: No

Reviewer #2: No

---

## [Editor Report · Acceptance letter]

7 Feb 2023

PONE-D-22-29069R1 

AA-amyloidosis in cats (*Felis catus*) housed in shelters 

Dear Dr. Ferri:

I'm pleased to inform you that your manuscript has been deemed suitable for publication in PLOS ONE. Congratulations! Your manuscript is now with our production department. 

Kind regards, 

on behalf of

Dr. Maria Stefania Latrofa 

Academic Editor

PLOS ONE